# SocialMedia2Traffic: Derivation of Traffic Information from Social Media Data

**Mohammed Zia** [1,†] **, Johannes Fürle** [2,†] **, Christina Ludwig** [2,†] **, Sven Lautenbach** [1,2,*] **, Stefan Gumbrich** [1] **and Alexander Zipf** [1,2]

1    HeiGIT gGmbH, Schloss-Wolfsbrunnenweg 33, 69118 Heidelberg, Germany
2    GIScience Research Group, Heidelberg University, Im Neuenheimer Feld 368, 69120 Heidelberg, Germany
*    Correspondence: sven.lautenbach@heigit.org; Tel.: +49-6221-533-491
†    These authors contributed equally to this work.

**Abstract:** Traffic prediction is a topic of increasing importance for research and applications in the domain of routing and navigation. Unfortunately, open data are rarely available for this purpose. To overcome this, the authors explored the possibility of using geo-tagged social media data (Twitter), land-use and land-cover point of interest data (from OpenStreetMap) and an adapted betweenness centrality measure as feature spaces to predict the traffic congestion of eleven world cities. The presented framework and workflow are termed as SocialMedia2Traffic. Traffic congestion was predicted at four tile spatial resolutions and compared with Uber Movement data. The overall precision of the forecast for highly traffic-congested regions was approximately 81%. Different data processing steps including ways to aggregate data points, different proxies and machine learning approaches were compared. The lack of a universal definition on a global scale to classify road segments by speed bins into different traffic congestion classes has been identified to be a major limitation of the transferability of the framework. Overall, SocialMedia2Traffic further improves the usability of the tested feature space for traffic prediction. A further benefit is the agnostic nature of the social media platform's approach.

**Keywords:** vehicle traffic; social media; Twitter; OpenStreetMap; Uber Movement; classification; traffic prediction; machine learning

## 1. Introduction

The ability to monitor, analyse, visualise and predict urban traffic movement has increased in the past years because of the sudden increase in GNSS (global navigation satellite system)-enabled devices, such as smartphones [1,2]. The GNSS-enabled fleet of delivery vans, for example, is one such data source which makes it possible to monitor near-real-time traffic congestion in urban cities. Such data have increased the popularity of traffic mapping in quantitative mode among research groups [3–5]. Vehicle traffic in particular has been extensively studied using such data due to its direct impact on economy and livelihood. However, existing studies have always been limited to small study areas due to the scarcity of data coverage, scalable proxies and ground truth validation [6]. All major traffic data providers are currently either private companies or government bodies. Therefore, data are in general not publicly available for science and research—researchers depend on the goodwill of the organisations owning the data or on sufficient financial resources to obtain access to them. This is creating a growing necessity for cheap, publicly available, georeferenced datasets for traffic prediction with a considerable spatial and temporal resolution. Such datasets can also be used to support similar research questions related to $CO_2$ emission, urban planning, sustainable goals and the overall well-being of urban dwellers [7].

Information technology giants such as Google, Yandex, HERE, Mapbox, etc., primarily rely upon telemetry data to perform traffic estimation on a global scale [8,9]. Data capturing thus either takes place via their API (Application Programming Interface), via SDKs

(Software Development Kit) installed on various user mobile devices or through some third-party providers such as fleets of GNSS-enabled delivery vans [8,10]. The amount of telemetry data collected is in the order of millions of data points, as reported by Mapbox in 2017, when the company collected around 20 million geo-tagged points to generate a global traffic map.

Within academia, a lot of regional studies have explored the usability of social media data to understand human mobility, vehicle navigation, smart cities and urban modelling [11]. These activities have focused on aspects such as vehicle and pedestrian traffic congestion or origin–destination (OD) matrix estimations [12]. However, most of the work has been based on using media data (text, images, videos and tags) associated with a given media platform. For vehicle traffic modelling, the majority of work is based on using the content of text messages of Twitter data for selected world cities [13]. The core of the approach is to look for key traffic-related words (in the case of Twitter) to perform a sentiment analysis to identify traffic conditions in the vicinity. It involves text mining and in some cases OD matrix generation. These kinds of approaches have limitations in terms of transferability to other regions with different spoken languages. Furthermore, the approaches cannot easily be transferred to other media platforms. Coffey [14] tried to find the relationship between BikeShare data and the kinds of tweet messages made by users during trips. The underlying approach was to convert the mobility data into a set of words for reclassification. Another recent work by Yao [9] tried to explore human mobility patterns by deriving spatial clusters from tweet posting coordinates. It has been shown that tweet-related features can largely improve traffic prediction for road segments where travel demand plays a crucial role in determining congestion [15–17]. Zhao [18] investigated different adapted centrality measures for estimating traffic flow. Besides modelling the network using primary and dual graphs, he also modelled the network based on groups of densely connected streets, similar to a community-based road network model. He furthermore investigated the influence of network models on traffic flow estimation.

Steiger [19] commented that "the user-generated, textual content of tweets is noisy, making it challenging to apply natural language processing (NLP) techniques to identify meaningful information". He identified a few frequently repeating, daily patterns with similar time-dependent disruption characteristics along major arterial (ring) roads (similar to road centrality), as a proxy indicator of mobility behaviour. Gao [20] interestingly counterargued the usability of betweenness centrality for traffic prediction by reporting that it is not a stable indicator because of the variability in shape and range across study areas. To achieve more realistic OD trips, he used the distance–decay effect to give more weight to short rather than long distances. Another notable study on the subject came from Pun [21] and Giles [22], who modelled traffic flow based on three types of traffic flow information, i.e., annual average daily traffic (AADT), public traffic and private traffic. They further analysed how the OD population density affects the number of trips per route per day.

In this study, the possibility of using geo-tagged social media data along with an adapted and approximated road betweenness centrality measure and land-use land-cover point of interest (POI) feature spaces was explored as a potential proxy for vehicle traffic congestion. A global attempt has been made to better understand at which spatial and temporal resolution highly congested regions related to motorised vehicles can be predicted. In particular, the different modes of counting and quantifying data points and other identified proxies have been investigated at various scales, making this a proof-of-concept study. The authors present a framework to extract this information for different city street networks, named SocialMedia2Traffic (SM2T). The framework understands the upcoming surge of more publicly available social media datasets, thus investigating if they all could collectively be used to mimic the telemetry data of other paid solutions. We argue that the location-based analysis for traffic congestion is more scalable and robust as it does not have to rely on human sentiment detection. We propose a context-free way to extract traffic information. The approach aims to ensure that any future service based on its forecasts

will not have to go offline in the case that a specific data source goes out of service—as happened partially to the geo-tagged Twitter data in 2019 when the option was disabled to attached GNSS-derived coordinates to tweets, and only less precise location information could be added. The way media data are being used in SM2T is different from how the paid solutions are using counterpart telemetry data, as discussed in the following sections.

There are two ways to extract traffic information from human-generated social media data: (a) using the context of generated media, and (b) using the coordinate location of the user during the period of data generation [23]. The context-based approach is not transferable across different media platforms [24], and is complex but works with a limited amount of data. Location-based traffic retrieval, on the other hand, is scalable and easier to implement in the sense that it remains independent of any specific platform and shows more robustness with a strong proxy if provided in a sufficient amount [25,26]. How many data points are "enough" for location-based analysis for a given resolution presumably varies from region to region. Works such as that by Wang [27] present a similar framework to perform traffic mapping; however, this framework does not use social media data as a proxy and relies solely on GPS trajectories and POI data. Jayasinghe [28] investigated the suitability of centrality measures for estimating traffic flow. Similarly, Zhang [29] explored the relationship between land-use land-cover POIs and traffic flow congestion in multiple cities. Therefore, the literature supports the idea of using social media, POI and centrality data for traffic mapping, although there has been no combined effort to measure it all on a global scale. Furthermore, a framework is missing for the development of Software as a Service.

The objectives of this paper are as follows:

- To test the existence of a direct as well as combined relationship between geo-tagged Twitter data/land-use land-cover POI/betweenness centrality and high traffic congestion on roads.
- To propose a framework to train the model and argue its fitness for purpose.

There is no open web service available that provides a dynamic traffic congestion map on a global scale. One of the immediate users of the SM2T service would be publicly available routing services such as OpenRouteService [30], which would benefit in the form of a better Estimated Time of Arrival. Such a service is supposed to help many sectors such as courier delivery, urban planning, environmental modelling, carbon emission, etc. [31–34]. The Uber Movement speed dataset is available for obtaining the ground truth for selected world cities (data can be retrieved from Uber Movement, (c) 2021 Uber Technologies, Inc., San Francisco, CA, USA , [35]). No peer-reviewed work of this nature is available online on a global scale. The framework presented here aims to provide data on high traffic congestion on a global scale, in contrast to small test areas that current research studies already provide. Although the precision and spatial/temporal resolution of SM2T falls well behind any proprietary solution, this should be treated as a preliminary study to explore the possibility in this direction.

We explored the relationship between Twitter data and traffic speed using the following experimental set-up:

- We tested two Twitter-based proxies as a predictor: the number of users on a road segment and the number of users within a vicinity.
- We used land-use and land-cover-related POIs as an additional predictor and investigated four different ways of aggregating POI information for a given tile.
- We used an adapted centrality betweenness measure as an additional predictor. Betweenness centrality was measured with respect to a different set of POIs.
- We tested four different spatial resolutions for a regular grid-based tessellation model.
- We investigated different thematic resolutions: (i) using continuous traffic speed information and (ii) using traffic speed information classified into up to three congestion levels.
- We tested the performance of five machine learning models.

## 2. Methods and Data

### 2.1. Conceptual Framework

Different social media platforms generate information in different formats and modes (cf. Table 1), making any kind of traffic volume estimation highly platform-dependent. For example, a context- or text-based model trained on Twitter data cannot be used for the image-based Flickr data. Thus, context-based analysis requires extensive media-specific machine learning that needs to incorporate the specifics of the platform. On the other hand, a model trained only on the location and timestamp data from Twitter can be applied to any other platform as long as it provides the corresponding information.

**Table 1.** Different social media data sources along with their availability and type of data. Columns show if the data source is publicly available or not and if it contains user location, along with the timestamp. * Twitter users can currently decide at which spatial granularity they provide location data or can decide to provide no geo-location information at all. These data are accessible through the Twitter researcher API.

| Social Media | Publicly Available | User Location | Time Stamp | Datatype |
|---|---|---|---|---|
| Twitter | Yes | Yes * | Yes | Multimedia message |
| Foursquare | Yes | Yes | Yes | Search, discover and rank POI |
| Snapchat | Yes | Yes | Yes | Multimedia message |
| Flickr | Yes | Yes | Yes | Image and video message |
| Facebook | No | No | Yes | Multimedia message |
| Instagram | No | No | Yes | Image and video message |

The position and connectivity of road segments in a city can be expected to be associated with the average traffic volume. Roads with higher importance are assumed to have more traffic on average than others. In addition, the traffic is hypothesised to depend on the attractiveness of the area. The authors have assumed that this could be captured using the density of specific POIs. A higher number of these POIs was hypothesised to be positively associated with a higher traffic volume. The attractiveness of an area was assumed to increase by those POIs even if the POIs are not directly reachable by car, e.g., if POIs are located in pedestrian zones. In addition to these static factors, the traffic volume is shaped by dynamic factors, which lead to regular and irregular patterns due to the time of the day, the day of the week, the presence of a holiday or school vacation or special events such as construction work, big sports events, demonstrations, etc. The authors hypothesise that these dynamic factors could be captured by social media activity on the Twitter platform. These activities and traffic volume could be linked directly or indirectly depending on if one filters for passengers (hopefully not drivers) tweeting during the ride or for all Twitter users present in a region. The latter is presumably linked to traffic volume with some time lag. A higher level of social media activities might both precede and follow high traffic volumes depending on the direction of the travel. If traffic volume is predicted at a temporally aggregated level, this presumably leads to better predictions as short-term fluctuations are averaged out. Higher traffic volumes are assumed to be related to traffic congestion if the capacity of the road segment is exceeded.

The authors have conceptualised five different temporal resolutions for current analysis (cf. Figure 1): *Live Traffic layer> Time Aggregated layer > Weekday Aggregated layer > Week Aggregated layer > OSM Speed Limit layer*. The *Live Traffic layer* holds traffic predictions made using the near-real-time social media data, along with other feature spaces. As soon as this particular layer becomes outdated it will update all underlying layers, except for the *OSM Speed Limit layer*. The *Time Aggregated layer* contains the traffic status of pre-defined time-bins to reflect office commute hours. The *Weekdays Aggregated layer* represents the traffic status aggregated over the last four weekdays—for example, by aggregating the traffic status of the region by aggregating the last four Mondays if the user is requesting Monday traffic information. Similarly, the *Week Aggregated layer* holds traffic value by

aggregating the data of the last seven days, considering all time-bins. Furthermore, finally, if for a given region no information is available from any of the above layers the *OSM Speed Limit layer* is used to return the highway maximum speed limit. This traffic layer stack thus would ensure that the user receives at least some information for the requested road segment, with the highest weighting given to the *Live Traffic layer* and descending weight towards the *OSM Speed Limit layer*. The overall traffic prediction would be a weighted average of all these layers.

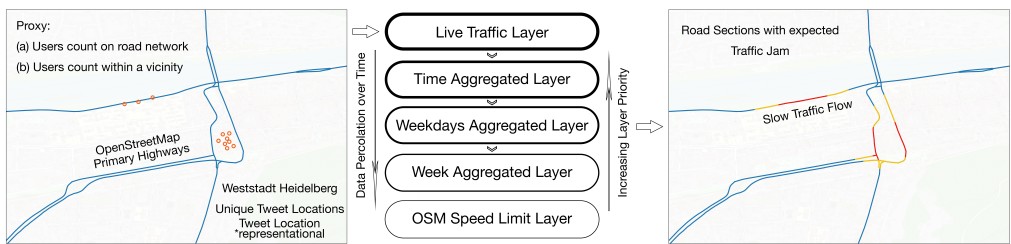

**Figure 1.** Conceptual layers contributing to the whole SM2T infrastructure. They are stacked in increasing order of priority. The *Live Traffic layer* is generated using two identified Twitter proxies, along with the land-use land-cover POI and betweenness centrality (cf. main text). Note that on the left map only Twitter proxies are shown.

A relatively high availability of social media at a relatively high spatial and temporal resolution was expected as a requirement for the reliable use of the SM2T traffic layer stack. Since the selected feature spaces directly affect only the *Live Traffic layer*, the rest of the article only explores the relationship between the two.

### 2.2. Cities Used for the Case Study

Eleven world cities (cf. Table 2) for which ground truth Uber speed data were available were selected to test the different models. The bounding boxes of the city per GADM admin level 2 boundaries [36] were used to select and clip the datasets.

**Table 2.** Cities used for the case study.

| Country | City |
| --- | --- |
| Brazil | Sao Paulo |
| Germany | Berlin |
| Kenya | Nairobi |
| Spain | Barcelona, Madrid |
| United Kingdom | London |
| Ukraine | Kyiv |
| USA | Cincinnati, New York City, San Francisco, Seattle |

### 2.3. Uber Movement Data

Uber is a mobility service provider, which has recently started providing urban vehicle speed data under the Creative Commons Attribution Non-Commercial License [35]. The authors used it to identify highly traffic-congested road segments in the selected cities. The data come with an OSM highway id as a foreign key that allows matching it to the corresponding OSM road network. Uber provides vehicle speed data for segments of OSM road objects. The road segments are defined by specifying the OSM-id of the start and the end of the segment. In addition, the driving direction is indicated. For the selected cities, average movement speeds differed, being between 27.0 km/h for Sao Paulo and 51.5 km/h for Cincinnati (cf. Table 3).

**Table 3.** Number of Twitter users, number of Tweets and average Uber speed for the cities used as case studies. Time period: March 2018 till November 2019.

|  | **Twitter Users** | **Tweets** | **Uber Average Speed (kph)** |
|---|---|---|---|
| Barcelona | 44,758 | 542,076 | 33.43 |
| Berlin | 26,070 | 418,932 | 38.97 |
| Cincinnati | 11,445 | 157,754 | 51.55 |
| Kyiv | 5398 | 73,046 | 40.36 |
| London | 151,509 | 1,543,018 | 34.10 |
| Madrid | 58,505 | 552,925 | 40.53 |
| Nairobi | 12,750 | 130,681 | 30.55 |
| New York City | 198,144 | 3,981,137 | 31.17 |
| San Francisco | 77,356 | 1,380,504 | 44.44 |
| Sao Paulo | 89,599 | 1,263,890 | 27.04 |
| Seattle | 34,694 | 518,950 | 46.72 |

The estimation of the traffic congestion class per tile was based on these vehicle speed data for March 2018 till November 2019. Each road segment was assigned to a traffic congestion class using Table 4. OSM road objects were intersected with the tiles. For each tile the weighted mean of the traffic congestion class was calculated. Thereby, the length of the split road segments was used as the weight. As the traffic congestion classes were coded as 0, 1 and 2, rounding the weighted mean to the nearest integer resulted in the estimated traffic congestion class (i.e., HTC, MTC or LTC). Road segments with missing Uber data were ignored.

**Table 4.** OSM highway speed bins, along with road width measures and buffer sizes used for tweet filtering by the two proxies: *User count within a vicinity* and *User count on a road segment*. HTC: High Traffic Congestion, MTC: Medium Traffic Congestion, LTC: Low Traffic Congestion.

| **Highway Type** | **HTC** | **MTC** | **LTC** | **Buffer in Vicinity** | **Buffer on Road** |
|---|---|---|---|---|---|
|  | **Speed bin (km/h)** | | | **(m)** | **(m)** |
| Motorway | 0.0–37.3 | 37.3–62.1 | >62.1 | 300 | 11.25 |
| Trunk | 0.0–37.3 | 37.3–62.1 | >62.1 | 150 | 11.25 |
| Primary | 0.0–24.8 | 24.8–43.5 | >43.5 | 150 | 7.00 |
| Secondary | 0.0–24.8 | 24.8–43.5 | >43.5 | 50 | 7.00 |
| Tertiary | 0.0–24.8 | 24.8–43.5 | >43.5 | 50 | 6.50 |
| Residential | 0.0–18.6 | 18.6–37.3 | >37.3 | 50 | 6.00 |

*2.4. Twitter Proxy*

To use the distance formula to generate traffic maps, one needs to have at least one vehicle on a given road segment for a specific point in time with a high frequency of data generation. In general, the SDKs of proprietary solutions generate GNSS data points every 10 s. Unlike telemetry data, geo-tagged social media data cannot be used in this manner because of the insufficient data coverage (Twitter provides only 1% of the global tweet coverage via its streaming API) and the innate nature by which it gets generated. A given user generating a social media feed at an interval of 10 s or less ought to be considered as an outlier or artefact.

There are two ways in which geolocation can be attached to tweets, if the user decides to do so: by the *coordinates* and by the *place* key. The *coordinates* key was used before 2019 to store GNSS coordinates from the device the user tweeted from. This is no longer possible as of 2019, at least, using the official Twitter client. Third party apps—such as Instagram—still allow the key to be filled. However, Instagram seems to be using the *coordinates* key for a different purpose, providing not the current location of the user but the coordinates from a user-specified fixed location [37]. Therefore, using the *coordinates* key from current tweets would presumably lead to very misleading results. This behaviour of Instagram has been reported prior to 2019 [37]. The *place* key allows the selection of the coordinates of

a Twitter-defined named place near the location where the tweet is posted. These places are provided with different levels of granularity: the categories are "country", "city" and "poi".

The authors had access to Twitter data for March 2018 till November 2019 from the Twitter streaming API. Tweets with the *GeoCoordinates* key with exact GNSS coordinates were used. The number of distinct Twitter users and the number of tweets differed by more than one order of magnitude between the selected cities (cf. Table 3). The low data availability of Twitter was addressed by the use of two proxies. The *User count on a road network* proxy is based on all tweets present within a given buffer of a road segment (cf. Figure 2a). The buffer size was chosen depending on the different OSM road classes (cf. Table 4). This proxy is intended to capture the subset of the population on the road that tweets during the ride, which might be a small share of all road users. While the road width of OSM road types is expected to differ in different countries, the same buffers are used for analysis across all selected cities. Another way to estimate the actual number of road users or vehicles on the road is to use the *User count within a vicinity* (cf. Figure 2b) as a proxy. Key public spaces (cf. Table 5) were used to select tweet clusters without a buffer.

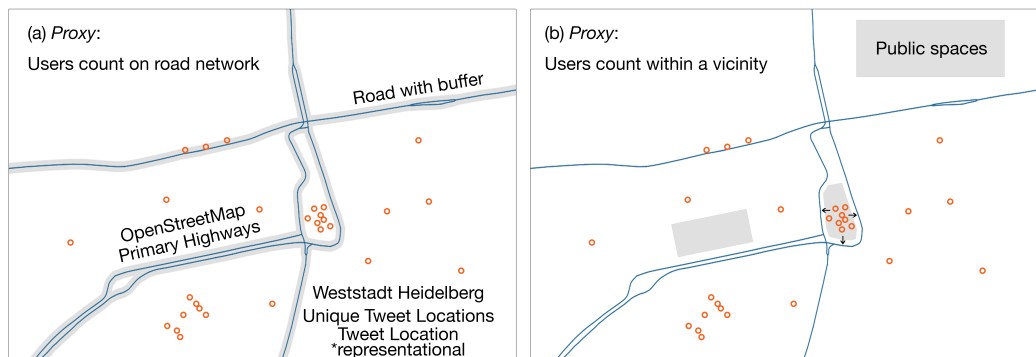

**Figure 2.** A sample street network in the city of Heidelberg, Germany, showing (**a**) how geo-tagged tweets were selected based on the buffer around the highways for the "User count on a road segment" proxy, and (**b**) how a geo-tagged tweet cluster in a public space was used for the "User count within a vicinity" proxy.

**Table 5.** Public space POIs used for the *User count within a vicinity* proxy as defined in OSM (key–value pair). The right column provides the values used for the different OSM keys of this proxy. These key–value pairs cover spaces where people gather during various hours of the day or on special events.

| Key | Value |
| --- | --- |
| amenity | parking, parking_space, marketplace |
| highway | rest_area, services, pedestrian |
| leisure | park, garden |
| landuse | recreation_ground, grass, village_green, cemetery, meadow |

The authors tested two additional approaches to extract the traffic information from Twitter data (cf. Table 6). The *direct speed estimation* approach uses data points as telemetry data using the speed formula. Its use directly mimics the telemetry data and the way they are used in proprietary solutions. If this approach could be used then there would be no need for any machine learning model in the *Live Traffic layer* and the use of other feature spaces. However, this approach proved not to be suitable for Twitter data due to the low frequency of data generation by the users—users rarely tweet several times while traversing a road segment. The *context-based* proxy has extensively been studied by researchers in the past. However, its high platform dependency and lack of standard key traffic phrases limit its transferability. The use of language-specific phrases and cultures makes context-based analysis cumbersome and highly complex. Data availability for a

context-based analysis is also low as only very few tweets contain traffic-related phrases or sentiments. Users tend to further post traffic-related tweets after an actual traffic jam, so this proxy could only be used for larger time-bins.

**Table 6.** Different types of proxies for geo-tagged Twitter data used in traffic modelling. Availability and priority should be high while complexity should be low for good results of traffic forecast modelling.

| Vehicle Speed Proxy | Availability | Complexity | Priority |
|---|---|---|---|
| Direct speed estimation | Low | Low | Very high |
| User count on a road segment | Medium | Medium | High |
| User count within a vicinity | High | High | Medium |
| Context-based | Low | Very high | Low |

### 2.5. Land-Use and Land-Cover Point of Interest Proxy

Land-use and land-cover POIs represent infrastructures that serve as a lifeline to any city. They range from supermarkets, hospitals, schools and universities to bus stops, car parks, etc. In SM2T, selected land-use land-cover POIs are used as an additional feature space for traffic prediction. The authors assigned the weights and radius of impact (cf. [38]) and used them during data processing to calculate their effects on the traffic of nearby road segments.

Four approaches to aggregate or count different land-use land-cover POIs near a road segment have been tested (cf. Figure 3). Aggregation was performed using a regular grid for tessellation. The idea is to use these data points to flag nearby road segments with various degrees of traffic congestion. The simplest approach (cf. Figure 3a) counts all POIs that are present in a given bounding box. These counts were then used to mark traffic flags, based on road type, for all road segments inside the bounding box. However, it was observed that different POIs could have different degrees of traffic impact based on their distance from nearby road segments and the number of people they serve. To capture this, road buffers and/or POI weighting were also used in order to filter and weight POIs (cf. Figure 3b–d).

### 2.6. Centrality Proxy

Finally, an adapted betweenness centrality measure of each road segment is used as an additional feature space. For SM2T, the authors adapted the original betweenness centrality measure [39] to include the spatial distribution of population density and POIs. This set of POIs is different from the land-use and land-cover POIs being used for the POI proxy in Section 2.5. Gao [40] showed that choosing the start and end points of simulated trips based on population density and POI data yields a closer relationship between the centrality indicator and the real traffic flow within a city. Major road segments that connect the majority of the population to critical infrastructure such as supermarkets, hospitals, etc. could be identified, and subsequently marked as more susceptible to traffic jams. The Ohsome API [41] was used to download the city's road network and selected POIs. We calculated centrality by simulating 20 k trips per city, themed and weighted by human action functions using the openrouteservice (https://openrouteservice.org accessed on 4 April 2022). The trips were simulated based on sampled pairs of locations for each city. One location was drawn based on an uneven sampling probability approach using population density as the probability weight. The other was selected from the human action function POIs. Population data were derived from the Global Human Settlement Population Layer [42,43], a raster dataset of 250 m cell size. The human action function POIs involved POIs from the functional groups "work", "education", "shopping" and "recreation". Trips shorter than 1km were rejected as we assumed car travel. We also used a distance–decay function to generate more short-distance trips than long-distance trips. The matching of the routes to the OSM street network to calculate the centrality was performed using the fast map matching developed by Canfast [44].

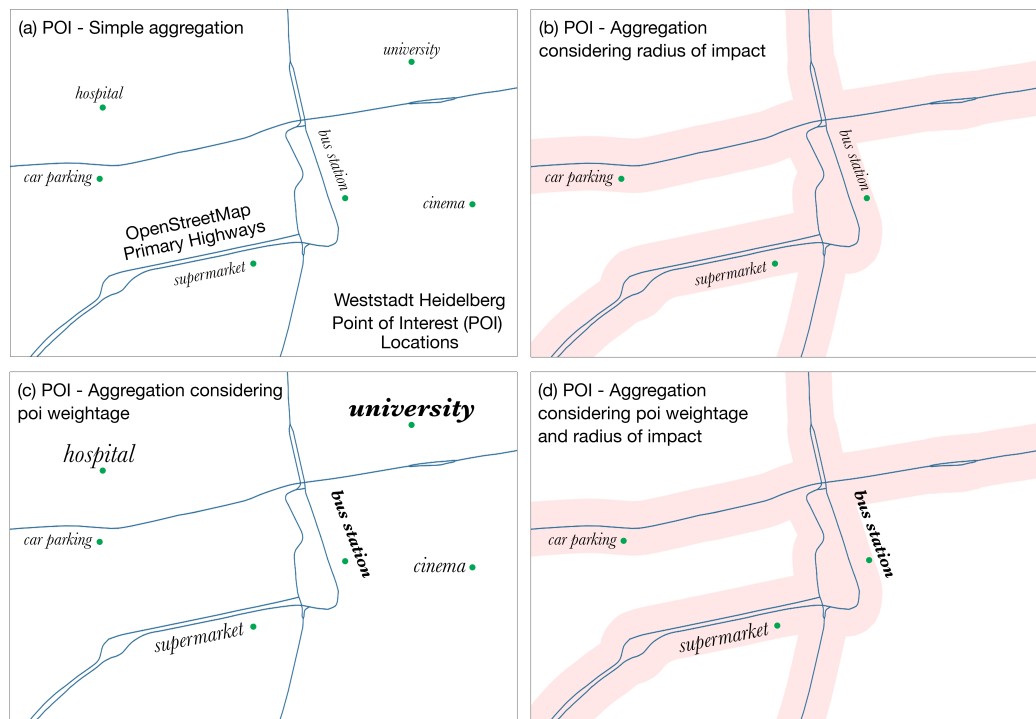

**Figure 3.** The figure shows four ways to aggregate land-use and land-cover POI data for a given tile. For (**a**), a simple counting of all POIs in an area of interest is performed. In (**b**) a pre-defined road buffer, per different highway type, is used to select only nearby POIs before counting. In (**c**) pre-defined weights, according to the importance of the infrastructure, are used while counting, and in (**d**) a pre-defined road buffer, per different highway type, in addition to weighting is used to select and prioritise only nearby POIs before counting.

### 2.7. Spatial, Temporal and Thematic Resolution

The speed at which vehicles are moving on a road segment—which is inversely proportional to the level of traffic congestion—is always a positive whole number. Ideally, the underlying problem should be viewed as a regression problem with all feature spaces and target labels being positive whole numbers. However, preliminary data visualisation and statistical analysis revealed that traffic predictions on a road level with a temporal resolution of a few minutes were not possible due to the very low availability of Twitter data. Among all considered independent variables, social media data were the only source that contained timestamp information. Thus, it was a controlling factor for temporal resolution for the *Live Traffic layer* (cf. Figure 1). Furthermore, the number of tweets map-matched to a given road segment was not sufficient to derive any statistically significant proxy at the scale of road segments for regression analysis. Therefore, the authors had to decrease the spatial, temporal and thematic resolution to make meaningful predictions. As for the spatial resolution, four tile sizes, i.e., 50, 100, 500 and 1000 m, were investigated to identify a good compromise between the resolution and accuracy of the prediction. As for the temporal resolution, initially, static time-bins representing office commuting hours were inspected (07:00–10:00 and 15:00–19:00); however, because of low number of data points per time-bin, the temporal variation of traffic was not considered in the model. Instead, the current study only investigated the overall nature of a given tile size in terms of traffic congestion with no time element.

To reduce the thematic resolution, the problem was converted into a classification problem by using the three traffic congestion classes as the final output: High Traffic Congestion (HTC), Medium Traffic Congestion (MTC) and Low Traffic Congestion (LTC). The definition of the three congestion classes for a given tile size is based on an equal quantile bin approach considering tweets, land-use land-cover POIs and betweenness

centrality indicators. The top 33% represents HTC, the bottom 33% LTC and the middle 33% MTC. Thereby, the authors were able to combine all OSM highway types, unlike for the regression, where each highway type had to be treated separately. This led to a higher number of training instances.

Finally, the Uber Movement data were used as the target label to train the model. Static speed bins (cf. Table 4) were used to convert the continuous Uber data to traffic classes for each OSM highway type (motorway, trunk, primary, secondary, tertiary and residential).

### 2.8. Missing Data Handling

After data processing, the dataset had multiple tiles (for each tile size) along with their respective HTC vs. no-HTC class labels for Uber, Twitter, land-use land-cover POIs and betweenness centrality. Since there were many tiles where at least one of the values was missing, the authors prepared three separate datasets to train the model. In the first dataset, only those tiles were taken where all non-null values for different columns were present. This dataset was the smallest in terms of the number of rows. In the second dataset, all those tiles were taken where the Twitter column had a non-null value. As for the cases where land-use land-cover POIs and/or the betweenness centrality class was missing, the no-HTC (dummy) label was used. Finally, in the third dataset, all tiles were taken and all empty cells were replaced with a no-HTC (dummy) label. The reason for using a no-HTC label as a dummy here was the assumption that the data might correctly show the absence of any tweet, thereby indicating low traffic congestion, and hence no-HTC. For all subsequent training, these three datasets were independently analysed.

### 2.9. Machine Learning Methods Comparison

To find the best-suited classification algorithm for logistic regression [45], Naive Bayes [45], k-nearest neighbours classifier [46], Random Forest [47] and Support Vector Machine models [48] were compared using the grid search CV approach in the Sklearn python package [49] for hyperparameter tuning. Although many other models could have been tested, these selected ones are the most standard models that are easy to implement and are also lightweight. Precision (cf. Equation (1)) was used for model evaluation as this was of highest priority for the project—recall or F1-score were not used.

$$\text{precision} = \frac{\text{TP}}{\text{TP} + \text{FP}} \tag{1}$$

with TP being true positives and FP being false positives.

The spatial unit at which the modelling happened was the level of the tiles.

### 3. Results

When comparing the prediction made by the centrality class as a predictor for different tile sizes and the three speed bins (HTC, MTC and LTC), the HTC class showed the highest accuracy (Figure 4). The comparison of the different tile sizes indicated that the relationship between the predictor and target variable was strongest at 100 and 50 m tile sizes (Figure 4). Similar conclusions were supported for Twitter proxies and the land-use land-cover POI predictors (results not shown). As the best prediction was achieved when predicting only HTC, the multi-class classification problem was subsequently converted to a boolean classification where HTC vs. no-HTC was evaluated and predicted.

Predictions based on the two proxies (*User count on a road segment* and the *User count within a vicinity*) were better than using the simple tweet count for high traffic congestion situations (cf. Figure 5). A similar result was found for the land-use land-cover POIs aggregation using the weight and radius of the impact method.

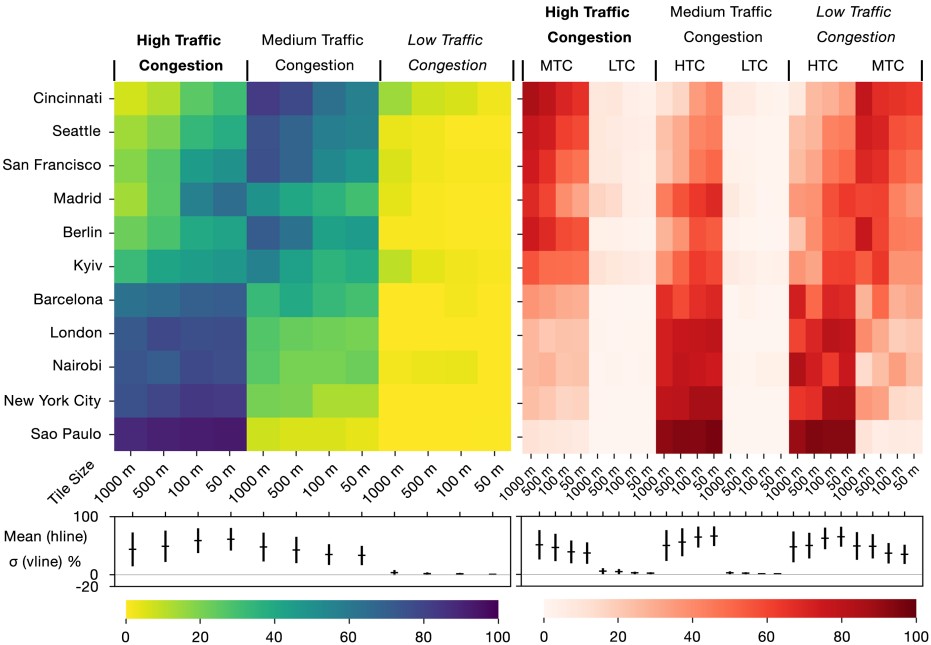

**Figure 4.** Accuracy of prediction of three traffic classes identified by the betweenness centrality measure, with the Uber Movement classes used as ground truth (**left**). Colours indicate the percentage of tiles correctly classified by the model using the adapted betweenness centrality as predictor. Extent of over/underestimation in wrongly classified classes using this proxy (**right**). Colours indicate the share of tiles incorrectly classified by the model using adapted betweenness centrality as a predictor.

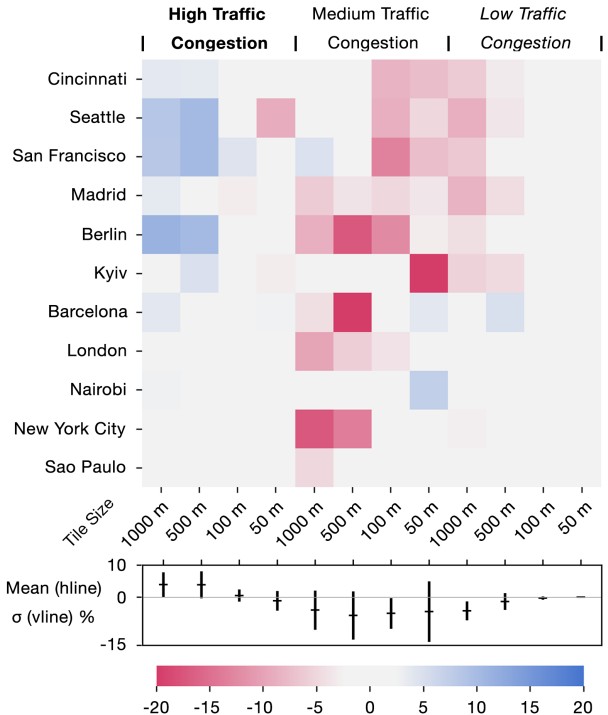

**Figure 5.** The plot compares the performance of two Twitter aggregation methods per tile: (i) *User count on a road segment* and (ii) *User count on a road segment + User count within a vicinity*. The colour indicates the difference in the percentage of correctly classified tiles between the predictions based on the two approaches. A positive value (blue) implies that the aggregation method using the combination of the two proxies is a better predictor.

While the prediction for high traffic congestion was almost balanced for the combination of all cities, the predictions for the individual cities were unbalanced (cf. Figures 6 and 7). Cincinnati was almost completely no-HTC-labelled and Sao Paulo was almost completely HTC-labelled for the 500 and 100 m tile size. Kyiv, on the other hand, showed a more balanced distribution of HTC and no-HTC tile locations at the 1000 m and also at the 500 m tile size. It is interesting to note that cities that showed more high traffic congestion if the Uber data had a higher prediction accuracy. A more realistic approach, therefore, would be to consider all tiles from different cities to cancel out the extreme effect of this imbalance, although all cities were also independently inspected for training.

Then, comparing the five different machine learning models in terms of precision across all cities, the k-nearest neighbours classifier performed best in terms of average precision for all tile sizes (cf. Figure 8). However, the difference in performance among different models was very small, especially between the k-nearest neighbours and the Random Forest classifier. Still, we selected the former for further analysis, due to its lower model complexity. This is beneficial if the model should be used as part of a web service, as lightweight models support fast prediction and hyperparameter re-adjustment. In a study on traffic congestion prediction using a different set of predictors, the Uber team also identified the k-nearest neighbours classifier as well as the best performing model [50]. Among the different tile sizes, 100 and 50m gave the highest average precision. Over all models, the average precision was highest if tiles with missing feature information were removed instead of using a dummy value for those tiles.

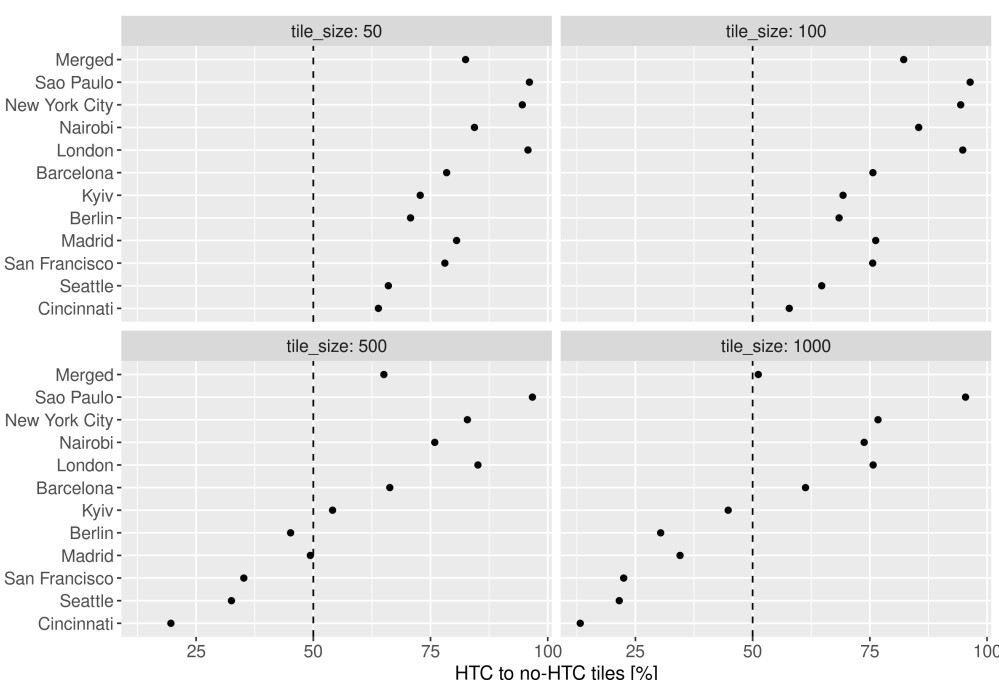

**Figure 6.** Degree of class imbalance in HTC vs. no-HTC labels per each dataset based on the Uber data (static class definitions). All instances with at least one empty feature space have been discarded. The x-axis shows the relationship between the number of HTC to no-HTC tiles per city for the different tile sizes. A value of 50% indicates perfectly balanced data. "Merged" represents the combination of all eleven cities.

The hypertuning of the k-nearest neighbours classifier suggested an optimal number of 30 neighbours for the 50, 100 and 500 m tile sizes (cf. Figure 9). For the 1000 m tile size, no clearly superior value could be identified. This is caused by the very low number of instances for such a coarse spatial resolution, which results in a very small number of neighbours, therefore not allowing for the proper optimisation of the parameter.

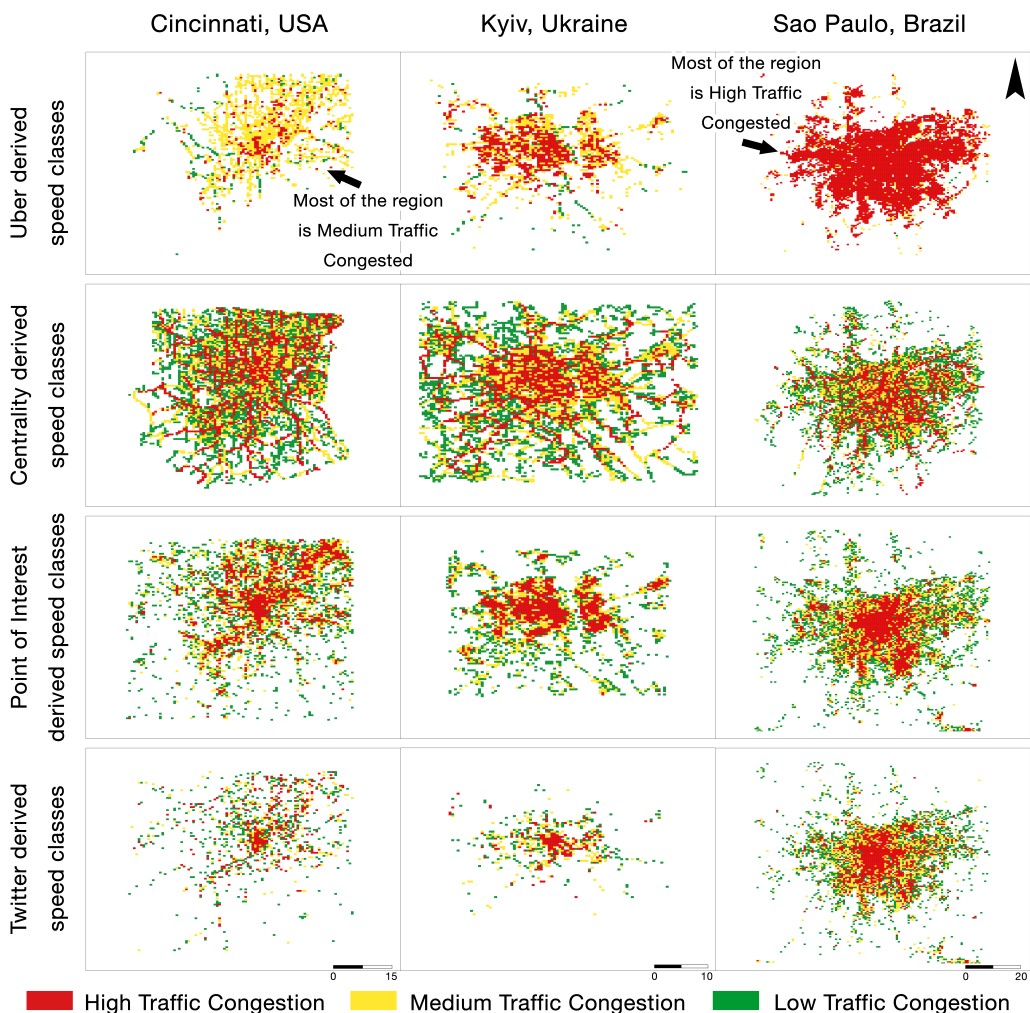

**Figure 7.** Visualising traffic congestion classes derived from all feature spaces using a quantile approach and validation (Uber) dataset for three cities (showing both edge case scenarios). The number of cells with predictions differs, as the predictors were not available for all tiles. The precision of the model predictions is presented in Figure 10.

For the 100 and 50 m tile sizes, the k-nearest neighbours classifier with 30 neighbours achieved more than 60% precision for each city (cf. Figure 10). For the combined dataset, the precision was above 80% for these two tile sizes. Although the precision was high, it is safe to expect it to be between 70% and 80% because the class imbalance for the combined dataset is still in a modestly imbalanced slab (cf. Figure 6). At the city level, precision was positively associated with the number of Twitter users (Pearson's r: 0.41) and with the number of tweets (Pearson's r: 0.37)—number of tweets and Twitter users per city were expectedly highly correlated (Pearson's r: 0.94). In addition, cities with a higher class imbalance were associated with higher precision (Pearson's r: 0.47).

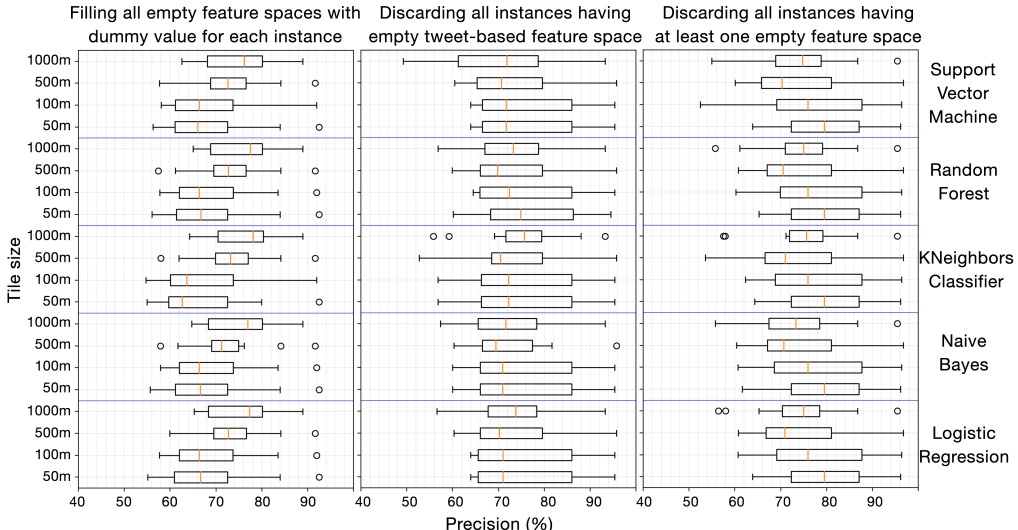

**Figure 8.** Comparison of the precision of five classification algorithms for different tile sizes. The three plots represent different datasets based on how empty cells were handled using the dummy value. The variability in the boxplots is due to the different precisions for the individual cities.

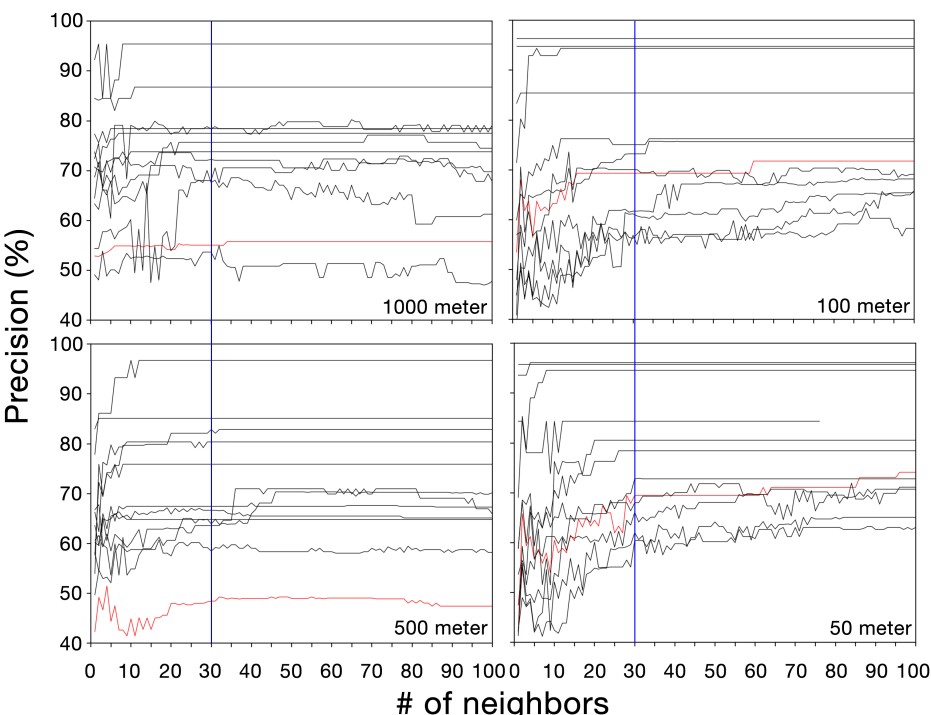

**Figure 9.** Selecting the best-performing number of neighbours of the k-nearest neighbours classifier for different tile sizes. The red curve represents the combined cities. The different black colours represent the individual cities. The vertical blue line characterises the selected value of 30 nearest neighbours.

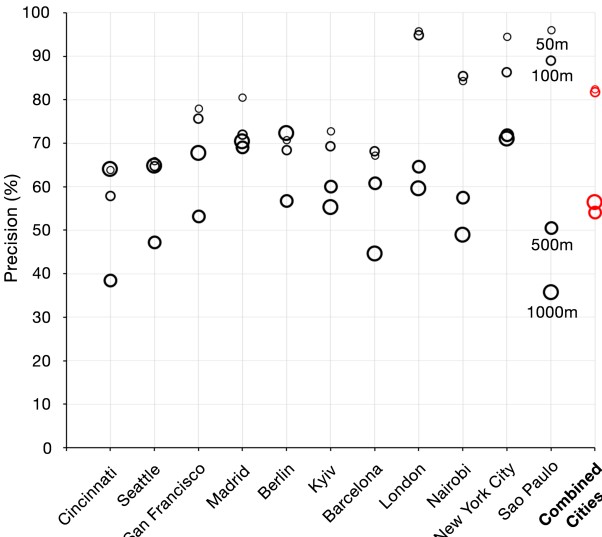

**Figure 10.** The performance of the k-nearest neighbours classifier (k = 30) for different tile sizes, different cities and a combined dataset. For each city, the model was trained using a 5-fold cross-validation approach using all cities but the selected city and validated against the latter. For the combined dataset, the model was trained using a 5-fold cross-validation approach and with precision calculated for the whole dataset.

## 4. Discussion and Limitation

The limited availability of georeferenced tweets was a major challenge for the prediction of traffic speed or traffic congestion. Therefore, we had to simplify the model and rely on proxies. The reduced model provides less information than originally intended, as it can only distinguish between HTC and no-HTC. This might be disappointing when comparing this approach with commercial approaches. Nevertheless, our approach captures the most important modes: stuck in traffic or moving. This provides important information for routing as HTC situations have serious travel time implications and highly effect the stress level of drivers [51]. Twitter discontinued its precise GNSS coordinate metadata service in 2019. Therefore, tweet availability is unlikely to improve. However, the proxy-based approach can easily be adapted to incorporate other publicly available geo-tagged social media data. Extending the proxies in that respect is likely to improve the quality of prediction.

The models were tuned for precision. The precision of 81% of the k-nearest neighbours classifier for the combined dataset at 100 m tile size indicates a good model performance. However, as the recall was not considered during the model selection, the prediction of the model might be too conservative, classifying road segments that are not suffering from congestion as highly traffic congested. Therefore, routing suggestions considering the model forecast presumably also tend to be conservative.

As traffic congestion prediction is performed at the level of tiles, the use of the prediction in routing adds additional uncertainty as the congestion affects all road segments that intersect with the tile. Traffic congestion on a highway might not affect traffic congestion at a nearby residential road. However, in many situations it is likely that traffic congestion is not limited to an individual road segment but affects the connected road segments as well.

Due to the lack of Uber data coverage, Asian and African cities were heavily under-represented in our case study dataset. Only one city, Nairobi, was available from these two continents. This raises concerns about transferring the model to Asian or African cities, which tend to be characterised by poorer OSM data quality and less precise population density information. For a few African or Asian cities, administrative or public domain datasets might be available that could be used instead of the Uber dataset. However, as these data come without reference to OSM objects, the use of these data would involve

map-matching between the data and the OSM road network. This was outside the scope of the current analysis but should be carried out in future studies.

To transfer the model to other cities, it will be necessary to calculate the centrality proxy for them as well. The simulation-based approach might become computationally demanding if the approach is transferred to a larger region, such as cities in the US or in India. Sufficient completeness of the road network is a prerequisite of the approach—while most cities are mapped relatively well [52], there are still cities for which completeness might be an issue. However, roads of higher importance tend to be mapped much more completely, so this might not be a real problem.

In addition, social media usage differs between countries. Differences affect the total number of social media users, the social media platform and the share of georeferenced tweets before 2019 [53]. This affects transferability, and the number of georeferenced posts needs to be sufficiently high. The number of twitter users and georeferenced tweets differs between the cities used as our case study (cf. Table 3). Model performance showed a positive association with these numbers, specifically for the smaller tile sizes. Hence, the transfer of the approach is presumably only promising for cities with a sufficient level of georeference social media activity. For cities with at least the same frequency of georeferenced social media posts as Kyiv (the lowest number of Twitter users and tweets in our case study sites), the approach might be applicable. However, this needs further testing.

For the classification of the Uber dataset to HTC and no-HTC labels, static class definitions (Table 4) were used. While this was necessary for a transferable approach on a global scale, it resulted in limitations in terms of the subjectiveness of the class meanings. A high, medium or low traffic speed limit is subjective terminology, which primarily depends upon the country, road type and the perspective of the commuter. Therefore, a more practical approach would be to define a global dictionary of what class definitions should be used for each city or country. However, to the best of the authors' knowledge, this has not been carried out by other studies so far, and further investigation in this direction was beyond the scope of the current study.

The individual cities differed strongly with respect to the amount of traffic congestion. While this reflects the range of traffic situations across cities, it might have impacted the quality of the model and the transferability to other cities. While the whole dataset seems relatively well balanced, individual cities contribute more or fewer labels for the HTC to the no-HTC class. Thereby, the characteristics of the extreme cases Sao Paulo and Cincinnati might have had a strong impact on what the model for all cities learned as important features for no-HTC and HTC prediction. Further studies in this respect seem necessary.

Another challenge for future research is the normalisation of the feature spaces across cities. The current percentile approach is purely data-driven and presumably is not transferable to cities outside of the case study. This might further lead to an artificial split between tiles of similar traffic conditions.

## 5. Conclusions and Future Work

Estimating traffic information from social media data is clearly challenging. Data availability has been indicated as a major obstacle in this regard. However, we have shown that it is possible to predict high traffic congestion when combining Twitter-derived proxy information with adapted network betweenness measures and land-use and land-cover POI information. We have demonstrated a reasonable way to use these proxies for the cities in the case studies. Our study highlights many of the challenges to be addressed and provides some guidance on reasonable compromises.

Prediction of high traffic congestion at the 100m tile level based on the presented approach are available via a web service (cf. Figure A1, SM2T https://sm2t.heigit.org accessed on 10 September 2022 ). This service has multiple modules in terms of API deployment, model training and hyperparameter tuning, allowing the sharing of results on a web-map and integration with third-party services such as the OpenRouteService. In

upcoming work, the authors will explore the feasibility of using the current model in Asian and African countries, by testing other social media sources and publicly available speed datasets as ground truth.

**Author Contributions:** Conceptualisation: Mohammed Zia, Johannes Fürle, Christina Ludwig, Stefan Gumbrich, Sven Lautenbach and Alexander Zipf; methodology: Mohammed Zia, Johannes Fürle, and Christina Ludwig; software: Mohammed Zia and Christina Ludwig; validation: Mohammed Zia and Christina Ludwig; data curation: Mohammed Zia and Johannes Fürle; writing—original draft preparation, all authors; writing—revision: Sven Lautenbach, Mohammed Zia and Christina Ludwig; visualisation: Mohammed Zia and Christina Ludwig; supervision: Alexander Zipf, Sven Lautenbach and Stefan Gumbrich; project administration: Mohammed Zia, Alexander Zipf and Stefan Gumbrich; funding acquisition: Mohammed Zia, Alexander Zipf, Sven Lautenbach and Stefan Gumbrich. All authors have read and agreed to the published version of the manuscript.

**Funding:** This research was funded by the German Federal Ministry for Digital and Transport (BMDV) in the context of the research initiative mFUND (grant numbers 19F2162A and 19F2162B). The authors gratefully acknowledge the data storage service SDS@hd supported by the Ministry of Science, Research and the Arts Baden-Württemberg (MWK) and the German Research Foundation (DFG) through grant INST 35/1314-1 FUGG and INST 35/1503-1 FUGG. Data were retrieved from Uber Movement, (c) 2022 Uber Technologies, Inc., https://movement.uber.com. Stefan Gumbrich and Sven Lautenbach acknowledge support by the Klaus Tschira Stiftung, Germany.

**Institutional Review Board Statement:** Not applicable.

**Informed Consent Statement:** Not applicable.

**Data Availability Statement:** Modelled traffic speed data are available for selected cities under https://github.com/GIScience/socialmedia2traffic-api (accessed on 26 June 2022)..

**Conflicts of Interest:** The authors declare no conflicts of interest.

## Abbreviations

The following abbreviations are used in this manuscript:

| | |
|---|---|
| AADT | Annual Average Daily Traffic |
| API | Application Programming Interface |
| GNSS | Global Navigation Satellite System |
| GPS | Global Positioning System |
| HTC | High Traffic Congestion |
| HTTP | Hyper Text Transfer Protocol |
| LTC | Low Traffic Congestion |
| ML | Machine Learning |
| MTC | Medium Traffic Congestion |
| NLP | Natural Language Processing |
| OD | Origin–Destination |
| OSM | OpenStreetMap |
| POI | Point of Interest |
| SDK | Software Development Kit |
| SM2T | SocialMedia2Traffic |
| VGI | Volunteered Geographic Information |

## Appendix A. SM2T Architecture and Interface

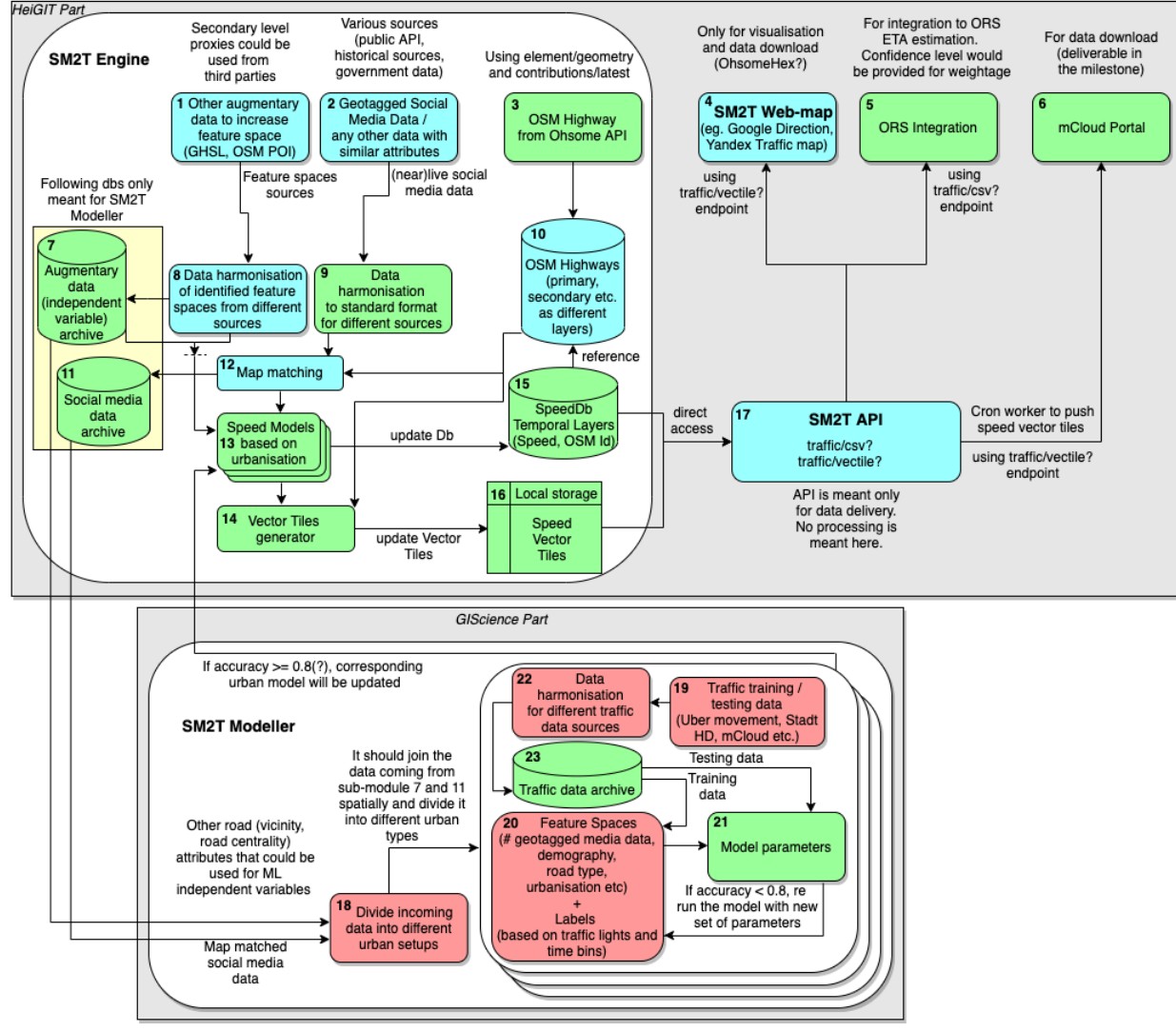

**Figure A1.** SM2T architecture and interface.

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
