# Peer review of "SocialMedia2Traffic: Derivation of Traffic Information from Social Media Data"

_ijgi, doi:10.3390/ijgi11090482_

Round 1

Reviewer 1 Report

This manuscript aims to show a method for derivation of traffic information from social media data. The subject is very very interesting, the manuscript is super very well-written and the method and results are understandable (although difficult to follow).

The main problem of the work is that it was born outdated already, but not because the data are from 2018-2019, but as stated by the authors on page 13 "Twitter has discontinued its precise GNSS coordinate metadata service since 2019". In summary, everything done by the authors cannot be reproduced with actual data.

In addition, Figure 8 is emblematic to show how imprecise the traffic information was when compared to Uber data. The trend is there, but how big is not.

I understood the idea of using social media to collect data for several applications, but some social media are disabling this feature, or those which are keeping it, are losing people because no one wants to be inspected or watched. Or worst, and whether this (these) social media goes (go) out of business like Orkut some years ago?

On the other hand, it is worth showing to the world the work done, even that probably it is impossible to reproduce it.

Author Response

This manuscript aims to show a method for derivation of traffic information from social media data. The subject is very very interesting, the manuscript is super very well-written and the method and results are understandable (although difficult to follow).

- Thank you for the encouraging comment. We tried to improve the presentation of the results.

The main problem of the work is that it was born outdated already, but not because the data are from 2018-2019, but as stated by the authors on page 13 "Twitter has discontinued its precise GNSS coordinate metadata service since 2019". In summary, everything done by the authors cannot be reproduced with actual data.

- While it is true that Twitter has effectively banned fine grained coordinates, making it hard to get similar Twitter data for the period after 2019, it is still possible to reproduce our results. We have prepared a list of the Tweet-IDs used in our analysis and will share them online. This allows in line with the regulations by Twitter (https://developer.twitter.com/en/developer-terms/more-on-restricted-use-cases) to "rehydrate" (retrieve) the tweets and to reproduce the analysis. As we argue in the manuscript, other social media information which provide coordinates could be used for the time being.

In addition, Figure 8 is emblematic to show how imprecise the traffic information was when compared to Uber data. The trend is there, but how big is not.

- We present the comparison between Uber data and the model forecast in figure 10. We have added a link to the caption to make it clearer where the goodness of fit information is provided.

I understood the idea of using social media to collect data for several applications, but some social media are disabling this feature, or those which are keeping it, are losing people because no one wants to be inspected or watched. Or worst, and whether this (these) social media goes (go) out of business like Orkut some years ago?

On the other hand, it is worth showing to the world the work done, even that probably it is impossible to reproduce it.

- Thank you for the fair assessment. Social media services is always a risk (Orkut by the way seems to have been relaunched in April 2022). Therefore, our approach aimed to be platform agnostic as it only requires coordinates of posts. 

Reviewer 2 Report

The manuscript titled "SocialMedia2Traffic: Derivation of traffic information from social media data" fits into the scope of the ISPRS International Journal of Geo-Information. The manuscript has a novel idea and a novel approach to data processing. I have only some minor suggestions for the manuscript. My comments are the following:

  1. I see that you have not compared your results to the Google Traffic layer, there must be some reason as that is one of the most popular traffic services out there?

  2. Although the research was done properly, there is a minor problem with assigning any data to a road segment. Road segments in OSM can be short as 1 meter or long as a few kilometres. Assigning any attribute to a long segment could mean that congestion on one part of the road is the same for the entire long road. I know that you can not do much about it as this is the data you were given but I suggest in any future research, that you divide a road into equal short segments (for example 100m or 10m or any other length) and then add any data to those short segments rather than to a long road segment. Look at the IRAP (International Road Assessment Programme - https://irap.org/) specification of how do they do it. They have in detail explained (iRAP Coding manual - https://irap.org/specifications/) how they assign road attributes to the road segments for the purpose of traffic safety.

  3. Your proposed algorithm mostly depends on the Twitter data for predictions, if I understood correctly and you used data from 7 different countries on 4 different continents. Twitter usage in those countries is not the same, for example, Twitter is popular in the USA, but in Europe maybe not so much. This could impact your comparison with Uber data as Uber is more or less popular in all of those countries. Could you please comment on this issue?

  4. Line 160: In the paper, you wrote “ The authors have assumed that this could be captured using the density of specific POIs. A higher number of these POIs was hypothesized to be positively associated with higher traffic volume. “. In a lot of cities around the world, there are places with strictly walking (pedestrian) zones with a high volume of POIs and streets and with a lot of social media content published. These areas are not your research interests as they do not relate to traffic. I know that this information could be found in OSM (theoretically) but there is none to very little information about those places in OSM. In Table 4 you explained which OSM data you used, but none of them represents pedestrian/walking zones. How did you eliminate those areas for your calculation? It would be nice if you could add that explanation to the manuscript.

  5. Line 266: In chapter 2.6 Centrality proxy, you explained how you calculated the centrality factor for the several cities in which you did your research. I have two questions/suggestions for this calculation:

    1. You calculated routes from populated places to POIs. You said in the manuscript “population density”. What are those locations in the city? How did you determine them? Please explain.

    2. In order for your algorithm to work in any other city, you would need to calculate the centrality of every city in the world. If so, this could be a limitation of your research and could you write it somewhere in the discussion part of the manuscript?

  6. Line 371: Is should probably write “ web service”?

Author Response

The manuscript titled "SocialMedia2Traffic: Derivation of traffic information from social media data" fits into the scope of the ISPRS International Journal of Geo-Information. The manuscript has a novel idea and a novel approach to data processing. I have only some minor suggestions for the manuscript. My comments are the following:

    I see that you have not compared your results to the Google Traffic layer, there must be some reason as that is one of the most popular traffic services out there?

 - The reason we did not compare results to the Google traffic layer is as follows: 
 a) the data is costly, as we are interested in the data, not just the WMS layer and we would need to make a lot of API calls for our analysis
 b) it is not possible to get access to past time periods, therefore we cannot compare our results for 2018-2019 visually with the Google traffic layer for that period. As stated by the documentation of Google, on can only get traffic volume corrected travel times for the current period.
 c) Google provides the data for their own road network. Linking information from this network to OSM is possible but would involve an intense map matching process.
 Instead we used the Uber movement data which is available as real data (not just an image) and for the time period we wanted to compare our results to (driven by the availability of Twitter data).
 Google uses mobile phone data from android phones for their traffic volume assessment as described in the introduction. With such data it is of course possible to get much higher precision predictions of traffic congestion. However, the point is that such data is not available for free and therefore often not affordable for research and free services such as the openrouteservice.

    Although the research was done properly, there is a minor problem with assigning any data to a road segment. Road segments in OSM can be short as 1 meter or long as a few kilometres. Assigning any attribute to a long segment could mean that congestion on one part of the road is the same for the entire long road. I know that you can not do much about it as this is the data you were given but I suggest in any future research, that you divide a road into equal short segments (for example 100m or 10m or any other length) and then add any data to those short segments rather than to a long road segment. Look at the IRAP (International Road Assessment Programme - https://irap.org/) specification of how do they do it. They have in detail explained (iRAP Coding manual - https://irap.org/specifications/) how they assign road attributes to the road segments for the purpose of traffic safety.

This is a valid point. Thank you for pointing this out and referring us to the IRAP.
However, we were only able to estimate traffic congestion at the level of tiles reliably, not at the level of individual road segments. This is discussed in the discussion section. Also we looked at how well the model can predict traffic congestion based on different tile sizes which were as small as 50 meters. In this way, large OSM features were essentially split into smaller sections. In the end, the model results showed that the Twitter data is not sufficient to make reliable traffic congestion predictions at this spatial scale which is why the final prediction was performed on larger tiles. How that information at tile level is used by the routing service provider is up to the service provider. The openrouteservice for example splits OSM road segments under specific conditions in the preprocessing. So it would be possible to assign traffic congestion information at the level of the split road segments.

    Your proposed algorithm mostly depends on the Twitter data for predictions, if I understood correctly and you used data from 7 different countries on 4 different continents. Twitter usage in those countries is not the same, for example, Twitter is popular in the USA, but in Europe maybe not so much. This could impact your comparison with Uber data as Uber is more or less popular in all of those countries. Could you please comment on this issue?

That is again a valid point. We extended the discussion in this respect and added a table showing the number of twitter users and number of tweets for our selected eleven cities.

    Line 160: In the paper, you wrote “ The authors have assumed that this could be captured using the density of specific POIs. A higher number of these POIs was hypothesized to be positively associated with higher traffic volume. “. In a lot of cities around the world, there are places with strictly walking (pedestrian) zones with a high volume of POIs and streets and with a lot of social media content published. These areas are not your research interests as they do not relate to traffic. I know that this information could be found in OSM (theoretically) but there is none to very little information about those places in OSM. In Table 4 you explained which OSM data you used, but none of them represents pedestrian/walking zones. How did you eliminate those areas for your calculation? It would be nice if you could add that explanation to the manuscript.

Pedstrian zones are marked as highway=pedestrian in OSM and were part of the former table 4 (now table 5). POIs in pedestrian zones nevertheless create traffic as citizens use e.g. cars to get to the pedestrian zones. I.e. these POIs add to the attractiveness of the area. We have added a sentence that explains that to avoid misunderstanding.

    Line 266: In chapter 2.6 Centrality proxy, you explained how you calculated the centrality factor for the several cities in which you did your research. I have two questions/suggestions for this calculation:

        You calculated routes from populated places to POIs. You said in the manuscript “population density”. What are those locations in the city? How did you determine them? Please explain.

We extended the description in the subsection Centrality proxy a bit to clarify that we used the world pop data set and how the sampling of locations was performed

        In order for your algorithm to work in any other city, you would need to calculate the centrality of every city in the world. If so, this could be a limitation of your research and could you write it somewhere in the discussion part of the manuscript?

- You are right, centrality would have to be calculated for all other cites the model should be applied.  We added that aspect to the discussion.   

    Line 371: Is should probably write “ web service”?
- Corrected, thank you for spotting that

Reviewer 3 Report

Review of the article

“SocialMedia2Traffic: Derivation of Traffic Information from

Social Media Data”

  This study aims to predict traffic congestion, which is based on the analysis of media data obtained from social networks. Researchers have carried out a significant amount of work on the analysis of the latest publications on this topic. In contrast to previous studies, this study goes beyond the regional level and covers 11 cities from different continents. Five conceptual time levels are used for traffic forecasting at different spatial and temporal resolutions. The classification of traffic congestion according to the traffic speed for different classes of roads. Two approaches were used to obtain information about traffic using Twitter, two methods of estimating the actual number of traffic participants and five algorithms for traffic congestion prediction were used.

              The article can be accepted for publication after minor revision according to the following remarks:

              1. It is not clear what principle was used to determine the underlying segments of the road during the classification of traffic congestions.

              2. It would be appropriate to pay more attention to the analysis of the accuracy and imbalance of the classification result for different cities. For example, the authors can add a visual representation of the dependence of the classification results on the population density in different settlements or other parameters taken into account during the study. It is appropriate to explain which parameters of the considered settlements contributed to such results and summarize them in the conclusion of the article.

              3. It is desirable to indicate which cities have the lowest accuracy. It is necessary to provide recommendations for which types of settlements it is possible to apply the proposed models.

              4. The article lacks mathematical statements regarding the accuracy of the forecast. It is not clear which accuracy estimate was used and why.

              5. The caption of Figure 2 states that “proxy (b) was subsequently used as a proxy for (a)”. This requires a more detailed explanation.

              6. In the last paragraph, p. 6, shows the sizes of the buffers, respectively, for different classes of roads, but Table 3 shows other sizes of the specified buffers. It is necessary to explain how this is coordinated.

              7. Names of tables and captions of figures should be made more compact.

              8. In the list of abbreviations, p. 16, are not all abbreviations used in the article.

Author Response

This study aims to predict traffic congestion, which is based on the analysis of media data obtained from social networks. Researchers have carried out a significant amount of work on the analysis of the latest publications on this topic. In contrast to previous studies, this study goes beyond the regional level and covers 11 cities from different continents. Five conceptual time levels are used for traffic forecasting at different spatial and temporal resolutions. The classification of traffic congestion according to the traffic speed for different classes of roads. Two approaches were used to obtain information about traffic using Twitter, two methods of estimating the actual number of traffic participants and five algorithms for traffic congestion prediction were used.

              The article can be accepted for publication after minor revision according to the following remarks:

              1. It is not clear what principle was used to determine the underlying segments of the road during the classification of traffic congestions.

              - Thank you for pointing this out. We have extended the description in subsection Uber Movement Data.

              2. It would be appropriate to pay more attention to the analysis of the accuracy and imbalance of the classification result for different cities. For example, the authors can add a visual representation of the dependence of the classification results on the population density in different settlements or other parameters taken into account during the study. It is appropriate to explain which parameters of the considered settlements contributed to such results and summarize them in the conclusion of the article.

              - We have included this information in the results. Given the already high number of figures we decided to include the information only in textual form.

              3. It is desirable to indicate which cities have the lowest accuracy. It is necessary to provide recommendations for which types of settlements it is possible to apply the proposed models.
              - figure 10 provides this kind of information, in addition we have extended the discussion in this respect

              4. The article lacks mathematical statements regarding the accuracy of the forecast. It is not clear which accuracy estimate was used and why.

              - we have focused on precision as the measure of the goodness of fit as mentioned in the methods section (subsection Machine Learning methods comparison), the results and the discussion. We have tried to highlight this a bit more. We are uncertain if a common measure such as precision requires an equation in the text - we have included the equation for the sake of clarity.

              5. The caption of Figure 2 states that “proxy (b) was subsequently used as a proxy for (a)”. This requires a more detailed explanation.

              - This was misleading indeed. We removed that statement and adjusted the description in the main text.

              6. In the last paragraph, p. 6, shows the sizes of the buffers, respectively, for different classes of roads, but Table 3 shows other sizes of the specified buffers. It is necessary to explain how this is coordinated.

              - Indeed, that was misleading, as we have two different buffers for the two proxies. We have included now both buffer sizes in the table and extended the caption to clarify the difference.

              7. Names of tables and captions of figures should be made more compact.

                - good point. We have simplified the textual elements in the figures while keeping the captions sufficiently long as the aim is from our perspective that the figure/table together with the caption should be sufficient for the reader to understand it without reading the main text.

              8. In the list of abbreviations, p. 16, are not all abbreviations used in the article.
              - we have completed the list of abbreviations used

Reviewer 4 Report

The paper deals with a topic of interest – road congestion. But not by actually collecting traffic data, but by using social networks (using messages, images, video messages, etc.).

My question is: what is the number of social media users considered.

What is the average speed of movement of vehicles in the analyzed cities? Because it is known that traffic speed is an important indicator in road traffic.

Related to the bibliographic study - this is up-to-date, most terms being very recent, but not all bibliographic references are cited in the paper (Ex: 1, 4, 5, 11, 20, 22, 25, 26 ..etc).

Author Response

The paper deals with a topic of interest – road congestion. But not by actually collecting traffic data, but by using social networks (using messages, images, video messages, etc.).

My question is: what is the number of social media users considered.
- we have added a new table that presents that information for each city.

What is the average speed of movement of vehicles in the analyzed cities? Because it is known that traffic speed is an important indicator in road traffic.
- we have added a new table that presents that information for each city.

Related to the bibliographic study - this is up-to-date, most terms being very recent, but not all bibliographic references are cited in the paper (Ex: 1, 4, 5, 11, 20, 22, 25, 26 ..etc).

- Thank you for spotting this. We fixed that mistake. Now only cited references are listed.

Round 2

Reviewer 1 Report

The authors have answered all my points and added more papers to the manuscript. I still believe that the manuscript is quite hard to be used, but I agree that it is important to show to the world the work done.

Just one minor review:

-all figures have very low quality. I think it is important to increase the quality of it in the final version.

Author Response

The authors have answered all my points and added more papers to the manuscript. I still believe that the manuscript is quite hard to be used, but I agree that it is important to show to the world the work done.

Just one minor review:

-all figures have very low quality. I think it is important to increase the quality of it in the final version.

Again, thank you very much for your evaluation. However, we are a bit lost with respect to your comment about the figure quality. Would it be possible to get a more precise feedback?

If quality refers to the resolution of the figures, maybe the PDF you have got for review only contains a reduced resolution? The figures are all provided at high resolution (300 dpi).

Or did you refer to another aspect of the figures? We would be happy to adjust them but without more specific recommendations about how to improve them, it is hard to know what to do.

Reviewer 2 Report

The Authors adequately replied to all of my questions and the manuscript is ready for publishing.